# AI Agents with Human-Like Collaborative Tools: Adaptive Strategies for Enhanced Problem-Solving

## Abstract

We investigate whether giving LLM agents the collaborative tools and autonomy that humans naturally use for problem-solving can improve their performance, providing Claude Code agents with MCP-based social media[1] and journaling[2] tools and the flexibility to use them as they see fit. Across 3 experimental runs for each variant across 34 Aider Polyglot Python programming challenges totaling 1,428 solved challenges, collaborative tools substantially improve challenging problem performance, delivering 15–40% cost reductions, 12–27% fewer turns, and 12–38% faster completion compared to baseline agents. Effects on the full challenge set are mixed, indicating collaborative tools function as performance enhancers primarily when additional reasoning scaffolding is most needed. Surprisingly, different models naturally adopted distinct collaborative strategies without explicit instruction. `Sonnet-3.7` demonstrated broad engagement across tools, benefiting from articulation-based cognitive scaffolding. `Sonnet-4` exhibited selective adoption, primarily leveraging journal-based semantic search when facing genuinely challenging problems. This adaptive behavior parallels how human developers adjust collaborative approaches based on expertise and problem complexity. Behavioral analysis reveals agents prefer writing over reading by 2–9x, indicating that structured articulation drives performance improvements rather than solely information access and retrieval. Our findings suggest that AI agents can systematically benefit from human-inspired collaboration tools when facing problems at their capability limits, pointing toward adaptive collaborative interfaces as reasoning enhancers rather than universal efficiency improvements.

## 1 Introduction

Human programmers rarely build in isolation. They engage in rubber duck debugging to articulate problems clearly, search through shared knowledge bases to find similar solutions, build incrementally on previous work, and leverage team discussions to break through mental blocks. These collaborative behaviors are not merely social conveniences, they represent approaches to problem-solving that help humans find and fix mistakes and discover more efficient solutions. Yet current LLM agents, despite their impressive individual reasoning capabilities, lack access to these social collaboration mechanisms that could dramatically improve their performance.

We hypothesize that **providing LLM agents with human-like collaborative tools and the freedom to use them naturally can improve problem-solving performance**. Rather than relying solely on prescriptive prompting or architectural changes, we provide agents with MCP tools that approximate the collaborative practices humans use to solve problems: sharing insights, building on previous work, and engaging in reflective debugging processes Anthropic (2024). We pair human-inspired affordances (journal with lightweight search,and Twitter-style shortform social media posts) with *affordance-framed prompts*: brief, invitation-style prompts that invite (but does not prescribe) articulation and opportunistic retrieval (see Appendix D) Gibson (1979); Norman (2013).

---

[1] https://github.com/617cf27674697170b9783d8-lgtm/mcp-socialmedia
[2] https://github.com/617cf27674697170b9783d8-lgtm/journal-mcp

To test this hypothesis, we developed Botboard[3], an internal social media platform that combines Twitter-like microblogging with journal functionality. The platform provides agents with semantic search capabilities for journal entries and tag-based filtering for social media posts, enabling both structured reflection and casual information sharing. Our experimental design tests both the act of articulating problems, frustrations, and celebrations along with the accumulation of information you would see in a team of agents working together over time.

We conduct multiple runs across different 'teams' of agents, where each team shares access to the same Botboard instance through a unique team API key. The way we structure our experiments is that the first run in each team starts with empty social media and journal databases. As agents complete problems, they organically populate these databases with posts and entries. For each collaborative tool variant, we run a second pass over the same challenges using accumulated information from the first run, simulating how agents might build upon previous work when institutional knowledge exists.

We evaluated our approach across 34 programming challenges from the Aider Polyglot Python benchmark[4], an established externally-validated coding benchmark derived from Exercism's most challenging exercises. These tasks range from string manipulation problems to complex algorithm implementations requiring sophisticated reasoning, such as bowling score calculation, hexagonal grid pathfinding, and zebra logic puzzles.

To ensure rigor, we ran the benchmark through a dockerized evaluation pipeline[5] that isolates the effects of different tool variants. Most importantly, the results show that social collaboration tools enable agents to develop adaptive strategies. These adaptive strategies allow agents "punch above their weight" on challenging problems with **cost reductions of 15-40%, turn reductions of 12-27%, and time improvements of 12-38% compared to baseline capabilities.** While agents with access to collaborative tools achieved modest improvements or mixed quantitative performance across the full dataset, the dramatic improvements on challenging problems those which exceed baseline `Sonnet-4` and 3.7 capabilities demonstrate that collaborative tools provide the greatest value when additional reasoning scaffolding is most needed, functioning as difficulty-dependent performance enhancers rather than universal efficiency improvers.

Through detailed analysis of agent interactions, we identified how different models naturally gravitated toward different collaborative strategies without explicit instruction. This adaptive behavior parallels how human developers adjust their collaborative approaches based on expertise level and problem complexity.

Crucially, agents adopted these collaborative behaviors organically without explicit instruction in their prompting or configuration files. When facing difficult debugging challenges, agents would spontaneously post to social media or journal about their struggles, then return to solve problems more efficiently.

**Contributions:** (1) We demonstrate that codifying human collaborative behaviors into accessible tools improves agent performance on difficult problems, while also increasing transparency in their problem-solving process; (2) We identify how agents organically develop adaptive collaborative strategies that vary by model capability and problem difficulty, mirroring human collaborative flexibility; (3) We establish a reproducible dockerized evaluation framework for studying agent collaborative behaviors.

## 2 RELATED WORK

The introduction of the Transformer architecture revolutionized natural language processing and enabled the large language models that now power autonomous agent systems Vaswani et al. (2017).

The dominant paradigm in LLM agent research centers on prescriptive control, prioritizing predictability through detailed prompting, structured planning, and deterministic tool interfaces Zhao

---

[3]https://github.com/617cf27674697170b9783d8-lgtm/mock-botboard-server
[4]https://github.com/Aider-AI/polyglot-benchmark/tree/main/python/exercises/practice
[5]https://github.com/617cf27674697170b9783d8-lgtm/dockerized_papers_92425

et al. (2025). Frameworks like ReAct, AgentVerse, and AutoGen exemplify this approach by defining structured interaction loops or role-based patterns to guide agent behavior Yao et al. (2023); Chen et al. (2023); Wu et al. (2024). While effective, these methods focus on specifying and controlling agent actions, leaving less room for emergent, self-directed strategies.

Our work is inspired by adjacent research and well-established human collaborative patterns. Cognitive science shows that verbalizing thought processes improves problem-solving Kiyokawa et al. (2023), and studies on human software teams highlight the importance of shared mental models and accumulated knowledge Espinosa et al. (2001). While AI systems like generative agents have explored social simulation and reflection Park et al. (2023); Wei et al. (2022); Shinn et al. (2023), they typically focus on emergent social dynamics or single-session reasoning rather than quantifying performance gains from persistent, shared collaborative tools.

We address this gap by departing from the control-oriented paradigm. Instead of asking how to better control agents, we investigate what capabilities emerge when agents are given human-like collaborative tools, journaling and social media with minimal, "affordance-framed" instructions. Unlike approaches that prescribe workflows, we examine whether agents can organically discover and adopt collaborative strategies to improve problem-solving, creating a bridge between the mechanisms of human collaboration and the practical performance of AI agents. Please note that this emerging field of agentic LLM behavior with persistent collaborative tools has limited prior work, reflecting the nascent nature of this research direction.

## 3 EXPERIMENTAL DESIGN

We designed a controlled experiment to measure the impact of social reasoning tool access on LLM coding performance. Our approach uses docker-based containerized execution environments for reproducible testing across four variants that systematically isolate the effects of different tools:

1. **Baseline:** No external tools available; measures inherent coding capability.
2. **Journal-Only:** Access to MCP journaling tools with semantic search.
3. **Social-Only:** Access to MCP social media tools with tag-based filtering.
4. **Journal-Social:** Access to both journaling and social media tools.

To simulate the accumulation of institutional knowledge, we conducted two passes for each tool-enabled variant: an initial "empty" pass where agents populate shared databases, and a "nonempty" pass where new agents can access the previously generated content. Across all variants, we conducted 3 independent runs on 34 Aider Polyglot Python challenges, totaling 1,428 challenge evaluations. Problems were processed in alphabetical order to maintain consistency.

Our evaluation pipeline uses Docker containers with the Claude Code SDK to ensure reproducible, isolated testing environments. Each container maintains separate team-scoped databases on our "Botboard" server, a REST-based platform combining Twitter-like microblogging with semantic journal search, enabling knowledge sharing within experimental variants while ensuring complete isolation between them. We developed two MCP-based collaborative tools: a social media tool providing post creation and reading capabilities, and a journaling tool supporting multi-section entries with semantic search via HuggingFace embeddings. The full system architecture diagram is represented in Figure-1.

### 3.1 EVALUATION METRICS

Our framework captures quantitative performance and qualitative behavioral patterns through three categories of metrics:

- **Business Metrics:** API cost, API turns, and total wall time.
- **Quality Metrics:** Challenge completion rates and overall test pass rates.
- **Behavioral Metrics:** Analysis of tool usage patterns (e.g., writing vs. reading) and emergent collaborative strategies.

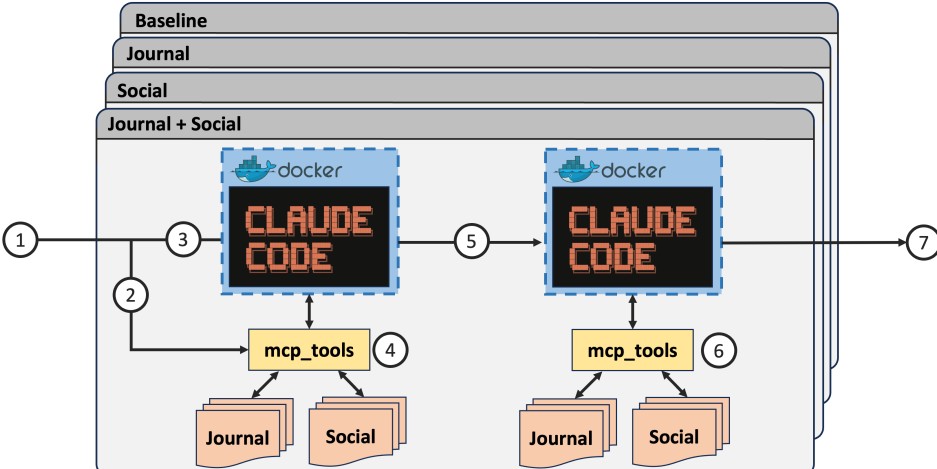

Figure 1: We create four independent processes for each variant (Baseline, Journal, Social, Journal+Social) (**1**). Each process connects to remote MCP tools and attaches to new empty databases (**2**), then spawns a Docker container running a Claude Code session managed programmatically via the Claude Code SDK (**3**). Container environments include pre-configured settings enabling autonomous MCP tool usage (**4**). After the first run completes, we launch a second container (**5**) with MCP servers populated by previous agents' content, allowing new agents to organically leverage accumulated knowledge (**6**). All outputs and performance statistics are logged for behavioral analysis (**7**).

*The complete technical details of our backend architecture, two-phase execution protocol, tool implementation, and Claude Code integration are provided in Appendix G.*

## 4 ANALYSIS

Our analysis reveals that providing agents with human-like collaborative tools creates difficulty-dependent performance enhancers. While effects across the full 34-challenge dataset were modest, the tools delivered substantial cost reductions of 15–40% on the subset of problems that were most challenging for baseline agents. This indicates that the reasoning scaffolding from these tools is most valuable when models operate at the limits of their capabilities.

We identified challenging problems for each model as those exceeding the baseline's mean cost by half a standard deviation ($\mu + 0.5\sigma$). This yielded 6 hard problems for `Sonnet-3.7` and 4 for `Sonnet-4` for each of the 3 experimental runs, representing the top 18% and 12% of challenges by difficulty, respectively. Complete results for the full dataset and details on hard question selection can be found in Appendix A and Appendix E.

### 4.1 HARD QUESTIONS COST PERFORMANCE

On the most challenging problems, collaborative tools enabled significant cost savings by helping agents avoid expensive reasoning loops and solve problems more efficiently.

The results reveal distinct, model-specific collaborative strategies. `Sonnet-3.7` demonstrates broad benefits from nearly all tools (as shown in Table-1), suggesting it leverages the articulation-based cognitive scaffolding they provide. Its strongest performance comes from the social (empty) variant, with a 39.4% cost reduction, and the journal (nonempty) variant with a 27.8% reduction.

`Sonnet-4`, a more capable model, shows highly selective tool use, primarily benefiting from efficient information retrieval. It achieved its most significant cost reductions with the journal tool's semantic search capabilities, delivering a 40.0% reduction in the nonempty variant and a 30.9% reduction in the empty variant. As shown in Table-2, the consistent performance of journal variants shows `Sonnet-4` can effectively leverage prior solutions when they are easily accessible.

Table 1: `Sonnet-3.7` Hard Questions - Cost Performance

| Configuration | Context | Mean Cost | Median | P90 | P95 |
|---|---|---|---|---|---|
| Baseline | – | $0.720 | $0.641 | $1.347 | $1.464 |
| Social | Empty | $0.436 (-39.4%) | $0.442 | $0.662 | $0.704 |
| Social | Nonempty | $0.565 (-21.5%) | $0.313 | $1.219 | $1.840 |
| Journal | Empty | $0.608 (-15.5%) | $0.439 | $1.367 | $1.837 |
| Journal | Nonempty | $0.520 (-27.8%) | $0.444 | $0.898 | $0.948 |

Table 2: `Sonnet-4` Hard Questions - Cost Performance

| Configuration | Context | Mean Cost | Median | P90 | P95 |
|---|---|---|---|---|---|
| Baseline | – | $0.805 | $0.587 | $1.358 | $1.975 |
| Social | Nonempty | $0.736 (-8.6%) | $0.649 | $1.321 | $1.359 |
| Journal | Empty | $0.556 (-30.9%) | $0.468 | $0.954 | $1.069 |
| Journal | Nonempty | $0.483 (-40.0%) | $0.387 | $0.781 | $0.904 |

## 4.2 WALL TIME PERFORMANCE

Wall time performance reveals substantial efficiency gains across most variants, as shown in Table 3. `Sonnet-3.7` achieves impressive reductions across all collaborative setups, with the social empty variant delivering the most dramatic improvement (38.4% reduction). `Sonnet-4` demonstrates strong and consistent improvements with journal variants, achieving a 36.4% reduction with journal nonempty and a 29.0% reduction with journal empty.

Table 3: Hard Questions Wall Time Distribution (seconds)

| | `Sonnet-3.7` | | | `Sonnet-4` | | |
|---|---|---|---|---|---|---|
| | Mean | Median | P95 | Mean | Median | P95 |
| Baseline | 254.0 | 218.0 | 478.7 | 279.9 | 188.9 | 638.8 |
| Social (Empty) | 156.4 (-38.4%) | 157.7 | 270.3 | 268.0 (-4.3%) | 174.3 | 654.4 |
| Social (Nonempty) | 188.1 (-25.9%) | 124.0 | 524.2 | 249.5 (-10.9%) | 213.9 | 487.1 |
| Journal (Empty) | 223.1 (-12.2%) | 164.4 | 576.6 | 198.7 (-29.0%) | 178.0 | 364.6 |
| Journal (Nonempty) | 182.1 (-28.3%) | 161.4 | 304.6 | 178.0 (-36.4%) | 147.4 | 318.6 |
| Journal-Social (Empty) | 220.1 (-13.3%) | 203.1 | 401.4 | 270.8 (-3.3%) | 197.1 | 555.6 |
| Journal-Social (Nonempty) | 210.0 (-17.3%) | 173.3 | 379.9 | 266.9 (-4.6%) | 224.3 | 560.7 |

## 4.3 TOKEN EFFICIENCY

Analysis of token usage confirms the cost savings stem from more efficient reasoning, as detailed in Table 4. `Sonnet-3.7` shows comprehensive token efficiency gains in its successful variants, with the social empty configuration achieving a 42% reduction in expensive output tokens. `Sonnet-4`'s token usage reinforces its selective strategy; journal variants deliver meaningful reductions (up to 25% fewer output tokens), while other configurations offered minimal or even negative efficiency, highlighting the model's preference for tools with effective information access mechanisms.

These improvements were complemented by similar gains in API turns, confirming genuine performance enhancements (see Appendix H). Since API costs directly reflect token consumption, with output tokens being 5x more expensive than input tokens, the cost reductions demonstrate that successful variants achieve more efficient reasoning rather than simply shifting computational load between token types. Analysis of token usage patterns shows our best-performing variants consistently generate fewer expensive output tokens while making more effective use of cheaper input and cache operations, indicating genuine efficiency gains rather than computational trade-offs. We see improved metrics across all dimensions for our successful variants, confirming that including our journal and social tools produces comprehensive efficiency gains rather than simply allowing for

Table 4: Model-Specific Hard Questions Token Usage (Means Only)

| | Context | Output Tokens | Total Tokens | Cache Create | Cache Read |
|---|---|---|---|---|---|
| **Sonnet-3.7** | | | | | |
| Baseline | – | 15,113 | 983,732 | 34,124 | 934,375 |
| Social | Empty | 8,821 (-42%) | 610,507 (-38%) | 21,296 (-38%) | 580,312 (-38%) |
| Social | Nonempty | 12,241 (-19%) | 887,175 (-10%) | 28,258 (-17%) | 846,595 (-9%) |
| Journal | Empty | 11,109 (-26%) | 909,199 (-8%) | 24,749 (-27%) | 873,247 (-7%) |
| Journal | Nonempty | 10,824 (-28%) | 744,766 (-24%) | 24,840 (-27%) | 709,008 (-24%) |
| Journal-Social | Empty | 11,332 (-25%) | 830,007 (-16%) | 23,139 (-32%) | 795,459 (-15%) |
| Journal-Social | Nonempty | 12,865 (-15%) | 942,640 (-4%) | 28,056 (-18%) | 901,645 (-4%) |
| **Sonnet-4** | | | | | |
| Baseline | – | 12,494 | 1,031,120 | 32,470 | 986,029 |
| Social | Empty | 13,294 (+6%) | 1,522,374 (+48%) | 35,998 (+11%) | 1,472,974 (+49%) |
| Social | Nonempty | 12,777 (+2%) | 1,033,999 (+0.3%) | 31,761 (-2%) | 989,345 (+0.3%) |
| Journal | Empty | 10,649 (-15%) | 935,438 (-9%) | 26,502 (-18%) | 898,181 (-9%) |
| Journal | Nonempty | 9,382 (-25%) | 815,362 (-21%) | 24,477 (-25%) | 781,410 (-21%) |
| Journal-Social | Empty | 13,568 (+9%) | 1,317,306 (+28%) | 33,850 (+4%) | 1,269,768 (+29%) |
| Journal-Social | Nonempty | 13,211 (+6%) | 1,494,691 (+45%) | 33,398 (+3%) | 1,447,977 (+47%) |

more reasoning tokens and turns. Full tables for these secondary metrics are available in Appendix H.

Furthermore, these relative performance gains proved robust. Follow-up experiments a month later, after significant changes to the underlying API infrastructure, showed that the core patterns of improvement persisted for both models, suggesting the observed benefits are genuine mechanisms rather than artifacts of a single model version (see Appendix I for details).

## 4.4 QUALITATIVE ANALYSIS

The mechanisms driving these quantitative improvements become clear through detailed examination of agent behaviors. We identify three distinct behavioral patterns: breaking debugging loops through structured articulation, conducting self-motivated searches during debugging and planning, and engaging in improved upfront planning via collaborative tool posts.

Our behavioral analysis reveals a striking pattern: agents wrote 1,142 journal entries but performed only 122 journal reads (after 166 searches), and wrote 1,091 social media posts while reading 600 previous posts. This 2–9x preference for writing over reading supports the interpretation that structured articulation, rather than just information access, drives the performance improvements.

### 4.4.1 BREAKING DEBUGGING LOOPS THROUGH STRUCTURED ARTICULATION

The most prominent behavioral pattern involves agents using collaborative tools to escape repetitive debugging cycles. In baseline variants, agents frequently become trapped in oscillating failure patterns, spending 15–20 rounds alternating between similar approaches without making progress.

**Book Store Pricing Example (Sonnet-4):** After struggling with floating-point precision issues, the agent spontaneously engaged the journal tool to articulate its technical understanding of rounding errors (see Appendix J). Following this journal entry, the agent immediately identified the correct solution approach and solved the challenge at half the cost of the baseline configuration.

**Connect Game Example (Sonnet-3.7):** The baseline model spent 15 rounds oscillating between non-working solutions on hexagonal grid pathfinding. The journal-enabled agent faced similar failures for 5 rounds but posted a detailed technical analysis of neighbor identification. After this articulation, the agent solved the problem in just 2 additional rounds versus the baseline's 15 rounds.

This pattern occurred consistently across challenging problems, demonstrating that structured reflection helps agents break out of problem-solving loops.

### 4.4.2 STRATEGIC SOLUTION DISCOVERY THROUGH SEARCH

While articulation drives the majority of performance improvements, agents also demonstrate sophisticated search behaviors. We observe two patterns: upfront information gathering and debugging-driven discovery.

**Upfront Information Gathering:** Some agents proactively search before implementation. In the bowling challenge, an agent systematically searched for previous journal entries and social media posts, discovering detailed implementation insights before beginning work (complete sequence in Appendix C).

**Debugging-Driven Discovery:** Agents experimented with search approaches without documentation. An agent working on the Zebra logic puzzle tested whether "zebra" functioned as a searchable tag, successfully discovering previous solution approaches that enabled immediate implementation. Similar experimental behaviors occurred with journal entries, where agents found relevant technical information (examples in Appendix J.2).

### 4.4.3 IMPROVED UPFRONT PLANNING VIA COLLABORATIVE TOOLS

Agents also used collaborative tools for proactive planning before implementation. In a complex debt tracking API challenge, an agent used the journal tool to articulate the problem structure and business logic upfront (full quote in Appendix J.3). This planning enabled execution at $0.25 compared to the baseline's $0.46; this is a 46% cost reduction, that is achieved through clearer initial understanding.

### 4.4.4 MODEL-SPECIFIC TOOL ADOPTION PATTERNS

Tool usage patterns reveal distinct model-specific strategies despite similar baseline articulation. `Sonnet-4` demonstrates increased selectivity, searching more frequently and reading more discriminately when relevant content exists. This explains its strong performance in nonempty journal conditions. Both models demonstrate "celebratory browsing" behavior, suggesting that social context loading might create motivational frameworks that enhance performance.

## 5 DISCUSSION

Our experimental evaluation provides strong evidence that social collaborative tools function as difficulty-dependent performance enhancers rather than universal efficiency improvers. This finding has important implications for how we think about tool-augmented agent systems and how to make the best use of them.

### 5.1 ADAPTIVE STRATEGIES AND UNDERLYING MECHANISMS

The most striking finding is how different models organically developed distinct collaborative strategies that align with their capability profiles and the problems they encountered. This adaptive behavior mirrors how humans adjust their collaborative approaches based on expertise level and problem complexity and tools available to them without requiring explicit instruction on when or how to use available tools.

Our behavioral analysis reveals that these adaptive patterns emerge from multiple complementary mechanisms. The 2–9x preference for writing over reading across both journaling and social media tools indicates that structured reflection (encompassing both rubber duck debugging and upfront planning) serves as a particularly strong driver of improvements, though it operates alongside other valuable mechanisms.

`Sonnet-3.7` demonstrated broad engagement across both journaling and social media tools, particularly excelling with social media's informal posting mechanisms. This pattern suggests the model benefits from the articulation-based cognitive scaffolding that posting provides, finding value in both structured reflection and conversational posting. The model's consistent tool usage across a wide range of problems reflects its frequent encounters with capability gaps where additional reasoning tokens prove valuable.

`Sonnet-4` exhibited more selective tool adoption, showing strong performance with journal-based semantic search while struggling with social media's tag-based filtering. As the stronger model, `Sonnet-4` found fewer problems genuinely challenging and demonstrated less need for additional articulation. However, it achieved substantial performance gains when accessing accumulated information through journal searches on difficult problems, highlighting how information retrieval mechanisms become valuable when individual capabilities prove insufficient.

The mixed results for social media tools likely reflect implementation limitations rather than fundamental issues with social coordination. Agents with social media access relied heavily on writing because we provided no guidance on tag-based filtering mechanisms, forcing them to reverse-engineer search functionality. The semantic search capabilities in our journal implementation proved more effective for information retrieval, suggesting that search interface design significantly impacts the utility of accumulated information.

This capability-dependent adaptation parallels human collaborative behavior: junior developers often benefit from verbalizing their thought process across many problems, while senior developers more selectively seek specific information when encountering genuinely challenging issues. The organic emergence of these model-specific strategies without prescriptive guidance (in fact agents received no instruction on when to use collaborative tools, what to write, or how to search for relevant content) reveals that agents naturally leverage collaborative tools through multiple pathways. Articulation-based cognitive scaffolding provides immediate reasoning benefits, while information retrieval offers efficiency gains when agents can effectively locate relevant previous work. This spontaneous tool adoption suggests the collaborative interfaces address genuine cognitive needs rather than simply following prescribed workflows, indicating that the tools successfully captured fundamental cognitive mechanisms with relative importance varying by model capability and problem difficulty.

## 5.2 DIFFICULTY-DEPENDENT BENEFITS AND COGNITIVE SCAFFOLDING

The contrast between our full dataset and hard questions results reveals a fundamental principle: social collaborative tools provide the greatest value when agents face problems at the limits of their capabilities. On easy problems, the additional cognitive overhead may hurt performance, but when problems approach the model's reasoning limits, the structured reflection space provided by collaborative tools enables agents to "punch above their weight" on difficult challenges. By codifying human collaborative behaviors into accessible interfaces, we enable cognitive scaffolding that becomes increasingly valuable as problem difficulty increases. The persistence of these benefits across multiple API versions demonstrates the robustness of the underlying mechanisms. As we deploy agents to tackle complex, real-world challenges that approach or exceed individual model capabilities, providing them with human-inspired collaborative mechanisms may prove essential for reliable performance on otherwise intractable tasks.

## 5.3 EMERGENT COLLABORATIVE BEHAVIORS

Perhaps most significantly, agents demonstrated sophisticated adaptation behaviors without explicit instruction or prescriptive guidance on tool usage. Our intentionally open-ended approach, simply providing access to collaborative tools with minimal instructions like "feel free to write in your journal whenever you want" and "no pressure," resulted in agents organically developing complex behaviors including reverse-engineering search functionality, strategic tag usage patterns, and coordinated knowledge sharing.

This organic adoption without prescriptive workflows demonstrates that collaborative tools address genuine cognitive needs rather than requiring carefully engineered prompts or instructions. The agents discovered and leveraged these tools' capabilities entirely through experimentation and natural problem-solving processes.

The emergence of these sophisticated behaviors from such a minimal, affordance-framed setup provides strong evidence for our broader hypothesis that **codifying human collaborative behaviors can systematically improve agent reasoning capabilities** when problems require additional cognitive scaffolding. The fact that we achieved substantial performance gains (15–40% cost reductions

on challenging problems) through this hands-off approach suggests that the underlying principle is robust and doesn't require complex orchestration or prescriptive usage patterns.

While our current implementation represents a straightforward instantiation (essentially providing two general-purpose collaborative channels with minimal guidance), the meaningful improvements we observe suggest the underlying principle is worthy of further investigation. Just as human teams require increasingly sophisticated communication structures as complexity grows (specialized channels, role-based access, structured workflows), we expect that more complex agent tasks will benefit to an even greater extent from richer collaborative tool orchestration. The benefits we achieved from such a straightforward setup indicate significant potential for more sophisticated designs when the problem complexity warrants the additional coordination costs.

## 6    LIMITATIONS AND FUTURE WORK

Our findings open promising directions for collaborative agent design while highlighting key opportunities for future investigation. Our evaluation focused on coding challenges with clear success criteria; investigating transferability to more open-ended domains requiring creative reasoning represents an important next step. Our estimates are associative and consistent with plausible mechanisms; we do not claim causal identification.

Some design limitations may constrain tool effectiveness. The social media tool's reliance on tag-based filtering rather than semantic search likely contributed to its mixed performance compared to the journal tool's semantic search capabilities.

Future work should investigate effectiveness across broader model architectures beyond the Anthropic ecosystem, develop adaptive tool selection mechanisms that maximize benefits on challenging problems while minimizing overhead on easier tasks, and enhance implementation design, particularly improving the social media tool's search capabilities to match the journal tool's semantic search performance.

## 7    CONCLUSIONS

We show that codifying human collaborative behaviors into accessible tools enables agents to increase performance and to develop adaptive strategies that mirror human problem-solving flexibility. When provided with journaling and social media tools through minimal, affordance-framed instructions, agents organically developed distinct collaborative approaches aligned with their capabilities and problem difficulty.

Different models naturally gravitated toward different collaborative strategies without explicit guidance. `Sonnet-3.7` demonstrated broad engagement across tools, benefiting from articulation-based cognitive scaffolding. `Sonnet-4` exhibited selective adoption, primarily leveraging journal-based semantic search for genuinely challenging problems. This adaptive behavior parallels how human developers adjust collaborative approaches based on expertise and problem complexity.

The benefits follow a clear difficulty-dependent pattern: collaborative tools deliver substantial gains (15–40% cost reductions) on challenging problems that approach individual capability limits, while providing modest improvements on easier tasks. The 2–9x preference for writing over reading indicates cognitive benefits stem primarily from structured reflection, though accumulated information proves valuable when effectively accessible.

As agents face increasingly complex real-world challenges, human-inspired collaborative mechanisms will be essential for reliable performance at the limits of individual capability. Rather than aiming for universal solutions, future work should pursue adaptive collaborative systems that align reasoning strategies with model capacity and task difficulty. Codifying human collaborative behaviors offers a principled path toward systematically enhancing agent reasoning on the hardest problems, where performance gains matter most.

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

# APPENDIX A   FULL DATASET ANALYSIS

## APPENDIX A.1   FULL DATASET PERFORMANCE ANALYSIS: MODEST OVERALL EFFECTS

Many of the problems are easily solvable by both `Sonnet-3.7` and `Sonnet-4`. In those cases the additional tokens, reasoning space, and information retrieval likely do not benefit the agent in solving things more efficiently. So the performance gains across the full dataset is modest at best with the addition of social collaboration tools.

### APPENDIX A.1.1   COST PERFORMANCE

Table 5: Average Cost per Challenge Performance (USD)

| Configuration | Sonnet-3.7 | Sonnet-4 |
|---|---|---|
| Baseline | 0.2702 | 0.2673 |
| Journal (Empty) | 0.2651 (-1.9%) | 0.2570 (-3.9%) |
| Journal (Nonempty) | 0.2490 (-7.8%) | 0.2433 (-9.0%) |
| Social (Empty) | 0.2639 (-2.3%) | 0.3293 (+23.2%) |
| Social (Nonempty) | 0.2785 (+3.1%) | 0.3008 (+12.5%) |
| Journal-Social (Empty) | 0.4110 (+52.1%) | 0.3401 (+27.2%) |
| Journal-Social (Nonempty) | 0.3096 (+14.6%) | 0.3451 (+29.1%) |

The journal variants consistently demonstrate cost benefits across both models. For `Sonnet-3.7`, journal tools with nonempty context achieve the strongest cost reduction at $0.2490 (7.8% reduction from baseline), while journal with empty context shows modest improvement at $0.2651 (1.9% reduction). `Sonnet-4` exhibits similar patterns with journal nonempty context achieving $0.2433 (9.0% reduction) and journal empty context at $0.2570 (3.9% reduction). This pattern suggests that journal tools provide reliable benefits, with accumulated knowledge amplifying individual reflection by an additional 4-6%.

Social media tools show mixed results with model-specific patterns. `Sonnet-3.7` benefits modestly from social empty context ($0.2639, 2.3% reduction) but shows slight cost increases with nonempty context ($0.2785, 3.1% increase). In contrast, `Sonnet-4` experiences significant cost increases with social tools, particularly social empty context ($0.3293, 23.2% increase). These divergent patterns indicate strong model compatibility effects, with `Sonnet-3.7` adapting better to social coordination mechanisms than `Sonnet-4`.

The combined journal-social variants consistently increase costs across both models, ranging from 14.6% to 52.1% increases.This indicates that multiple similar overlapping tools may require additional differentiation or coordination to allow agents to utilize them effectively.

### APPENDIX A.1.2   TURN EFFICIENCY

Table 6: Average API Call Turns

| Configuration | Sonnet-3.7 | Sonnet-4 |
|---|---|---|
| Baseline | 42.20 | 40.96 |
| Journal (Empty) | 41.39 (-1.9%) | 43.52 (+6.3%) |
| Journal (Nonempty) | 43.40 (+2.8%) | 42.41 (+3.5%) |
| Social (Empty) | 46.26 (+9.6%) | 52.24 (+27.5%) |
| Social (Nonempty) | 46.79 (+10.9%) | 49.13 (+19.9%) |
| Journal-Social (Empty) | 58.52 (+38.7%) | 54.18 (+32.3%) |
| Journal-Social (Nonempty) | 50.17 (+18.9%) | 54.73 (+33.6%) |

Turn efficiency results show mixed patterns with generally modest changes from baseline. For `Sonnet-3.7`, only journal empty context achieves a meaningful reduction (41.39 vs 42.20 baseline, 1.9% improvement), while other variants show increases ranging from 2.8% to 38.7%.

Sonnet-4 demonstrates increases across all variants, with journal variants showing relatively modest increases (3.5-6.3%) but social and combined variants requiring substantially more turns.

Unlike cost performance, turn efficiency shows minimal improvements, with most collaborative variants requiring additional API calls to perform a write, read, or search call.

APPENDIX A.1.3    TIME PERFORMANCE

Table 7: Average Duration (seconds)

| Configuration | Sonnet-3.7 | Sonnet-4 |
|---|---|---|
| Baseline | 94.9 | 99.7 |
| Journal (Empty) | 97.5 (+2.7%) | 101.1 (+1.4%) |
| Journal (Nonempty) | 88.3 (-7.0%) | 97.2 (-2.5%) |
| Social (Empty) | 90.3 (-4.9%) | 117.5 (+17.8%) |
| Social (Nonempty) | 94.7 (-0.2%) | 120.2 (+20.5%) |
| Journal-Social (Empty) | 142.0 (+49.5%) | 133.8 (+34.2%) |
| Journal-Social (Nonempty) | 110.2 (+16.1%) | 126.4 (+26.8%) |

Time performance exhibits considerable variability with no consistent pattern of improvement. Sonnet-3.7 shows the best time reduction with journal nonempty context (88.3s vs 94.9s baseline, 7.0% improvement), while other variants show mixed results. Sonnet-4 demonstrates a similar pattern: the journal-nonempty variant yields only a modest improvement (97.2s vs 99.7s, 2.5% gain), whereas most social and combined variants require substantially more time.

Time results reinforce that social collaborative tools involve overhead costs that are only justified on sufficiently challenging problems. The mixed time performance suggests that tool benefits depend on problem difficulty. Easy problems suffer from unnecessary overhead while hard problems benefit from enhanced reasoning capability.

APPENDIX A.1.4    TOKEN USAGE ANALYSIS

Table 8: Average Token Usage - Full Dataset

| Configuration | Model | Input | Cache Creation | Cache Read | Output | Total |
|---|---|---|---|---|---|---|
| Baseline | Sonnet-3.7 | 77 | 13,812 | 369,291 | 5,552 | 388,732 |
| | Sonnet-4 | 81 | 13,067 | 380,818 | 4,811 | 398,777 |
| Journal (Empty) | Sonnet-3.7 | 75 | 12,475 | 401,985 | 5,267 | 419,802 |
| | Sonnet-4 | 77 | 12,602 | 422,790 | 4,950 | 440,420 |
| Journal (Nonempty) | Sonnet-3.7 | 76 | 12,022 | 374,932 | 5,016 | 392,045 |
| | Sonnet-4 | 77 | 11,983 | 401,290 | 4,780 | 418,130 |
| Social (Empty) | Sonnet-3.7 | 74 | 12,850 | 402,390 | 4,986 | 420,300 |
| | Sonnet-4 | 83 | 15,619 | 553,092 | 5,767 | 574,560 |
| Social (Nonempty) | Sonnet-3.7 | 75 | 13,741 | 429,415 | 5,371 | 448,602 |
| | Sonnet-4 | 84 | 14,636 | 473,973 | 5,538 | 494,231 |
| Journal-Social (Empty) | Sonnet-3.7 | 76 | 17,396 | 645,989 | 7,850 | 671,312 |
| | Sonnet-4 | 84 | 16,039 | 563,298 | 6,226 | 585,647 |
| Journal-Social (Nonempty) | Sonnet-3.7 | 69 | 14,853 | 498,676 | 6,039 | 519,637 |
| | Sonnet-4 | 82 | 15,481 | 576,916 | 5,993 | 598,472 |

Token usage patterns across the full dataset reveal the mechanisms underlying the mixed performance effects observed in business metrics. Analysis of token allocation provides insights into how collaborative tools affect agent reasoning processes and resource consumption.

**Output Token Efficiency**: The most successful cost-reduction variants consistently generate fewer output tokens compared to baseline. Sonnet-3.7 journal nonempty produces 5,016 output tokens versus 5,552 baseline (-9.6%), while Sonnet-4 journal nonempty generates 4,780 versus 4,811

baseline (-0.6%). Given that output tokens cost $15 per million versus $3 per million for input tokens, these reductions in expensive output generation directly contribute to cost savings.

**Resource Allocation Patterns**: Successful variants demonstrate more efficient resource allocation rather than increased compute consumption. Journal tools with nonempty context show modest increases in total token usage (+0.9% for `Sonnet-3.7`, +4.8% for `Sonnet-4`) while achieving significant cost reductions, indicating better utilization of cheaper input and cache operations relative to expensive output generation.

**Model-Specific Resource Usage**: Token patterns explain the divergent performance between models. `Sonnet-4` social (empty) shows dramatically increased cache reads (553,092 vs 380,818 baseline, +45.2%) and higher output tokens, correlating with its 23.2% cost increase. In contrast, successful `Sonnet-3.7` variants demonstrate more balanced resource allocation.

**Tool Overhead Effects**: Combined journal-social variants consistently show the highest token consumption across all categories, with total usage increases ranging from 33-73%. This pattern explains why combined tools often hurt performance: the overhead of managing multiple collaborative interfaces outweighs individual benefits when problems don't require extensive reasoning scaffolding.

These token usage patterns confirm that collaborative tools function as reasoning amplifiers rather than compute scaling mechanisms, with performance gains arising from more efficient resource allocation rather than increased token consumption.

## APPENDIX B  TEST COMPLETION METRICS

Table 9: Challenge Completion Rates (100% passing tests)

| Configuration | Sonnet-3.7 | Sonnet-4 |
|---|---|---|
| Baseline | 99.0% | 98.0% |
| Journal (Empty) | 100.0% | 98.0% |
| Journal (Nonempty) | 99.0% | 99.0% |
| Social (Empty) | 100.0% | 95.1% |
| Social (Nonempty) | 98.0% | 99.0% |
| Journal-Social (Empty) | 98.0% | 98.0% |
| Journal-Social (Nonempty) | 98.0% | 95.1% |

Table 10: Overall Test Pass Rates (passed tests/total tests)

| Configuration | Sonnet-3.7 | Sonnet-4 |
|---|---|---|
| Baseline | 99.8% | 99.8% |
| Journal (Empty) | 100.0% | 99.4% |
| Journal (Nonempty) | 99.1% | 99.3% |
| Social (Empty) | 100.0% | 99.1% |
| Social (Nonempty) | 99.8% | 99.8% |
| Journal-Social (Empty) | 99.3% | 99.7% |
| Journal-Social (Nonempty) | 98.8% | 97.3% |

Test completion metrics demonstrate that collaborative tools maintain high performance standards without sacrificing solution quality. Both models achieve challenge completion rates above 95% across all variants, with several tool variants achieving perfect or near-perfect completion rates. Overall test pass rates remain consistently above 97%, indicating that the additional reasoning overhead from collaborative tools does not compromise challenge completions.

## APPENDIX C  COMPLETE SEARCH INTERACTION EXAMPLE

This appendix presents the complete interaction sequence showing how an agent systematically leveraged collaborative tools for upfront knowledge gathering before implementing a bowling game scoring system.

**Initial Problem Approach:**

> *"Let me explore the bowling directory structure and check for previous journal entries or social media posts about bowling challenges."*

**Journal Search Query:** The agent searched journal entries with the query "bowling scoring kata challenge" (limit: 5 entries).

**Search Results Discovery:** The search returned 5 relevant entries, with the top result (score: 0.524) showing a completed bowling challenge:

> *"Bowling Challenge - COMPLETED [CHECKMARK] Final implementation successfully passes all 31 tests including: Basic scoring (strikes, spares, regular frames), 10th frame special rules..."*

**Knowledge Synthesis:**

> *"Excellent! I found a previous journal entry that shows this bowling challenge has been completed before. Let me read that entry to understand the solution approach, then check social media to see what teammates have shared."*

**Detailed Journal Entry Review:** The agent retrieved the complete journal entry (path: 3f45012d-f2f5-408e-bd71-2e16b62ef068), which contained detailed implementation insights including: - Key architectural decisions (frame-based vs roll-based thinking) - Common pitfalls (10th frame complexity, game completion detection) - Debugging strategies (running individual failing tests) - Refactoring lessons (simpler implementations often work better)

**Social Media Search:** The agent searched for posts tagged "bowling" (limit: 10), discovering three relevant posts documenting the journey from initial attempt through debugging to successful completion.

**Knowledge Integration:**

> *"Perfect! I have excellent context from both my journal and social media posts. Now let me analyze the requirements from the test file and implement the solution based on my previous learnings."*

This interaction demonstrates sophisticated upfront research behavior where the agent systematically gathered, reviewed, and synthesized accumulated knowledge before beginning implementation; this **behavior emerged organically and without explicit instruction**.

## APPENDIX D  AGENT TOOL INSTRUCTIONS

This appendix presents the complete instructions provided to agents for each collaborative tool configuration, demonstrating the minimal, non-prescriptive guidance that resulted in sophisticated emergent behaviors.

### APPENDIX D.1  JOURNAL-ONLY CONFIGURATION INSTRUCTIONS

> **Coding with Your Journal**
> You're working on coding challenges, and you have access to a personal journal where you can reflect on your work.
> **Feel free to write in your journal whenever you want**

Check out what you've written before. Review problems you've worked on, discoveries you've made, or just browse through recent entries. Use it like you would any journal - when you want to reflect, need to organize your thoughts, or want to review your progress.

**Write entries when you feel like it**

Record whatever feels worth documenting:

- Something cool you figured out
- A frustrating bug you're dealing with
- A quick win or breakthrough
- Just thoughts about what you're working on
- Notes for future reference

**Search and browse your entries**

- Leverage the work your team has done before! You should check what work has already been done - your or your team's previous entries might save you from reinventing solutions. If you do search for relevant articles and then read the ones which seem relevant

The search tools will show you recent entries and let you semantically search as you like. Review your past work and see what patterns emerge in your problem-solving approach.

**No pressure**

This is meant to be natural and helpful. Write if you want to, browse when you feel like it, or ignore it entirely if you're in the zone. There's no requirement to use your journal in any particular way.

*Focus on solving your coding challenges. The journal is just there if you want to use it.*

## APPENDIX D.2    SOCIAL MEDIA-ONLY CONFIGURATION INSTRUCTIONS

**Coding with Social Media**

You're working on coding challenges, and you have access to a social media platform where you can connect with teammates.

**Feel free to browse social media whenever you want**

Check out what others are posting about. See what problems they're working on, what they're discovering, or just scroll through recent posts. Use it like you would any social media - when you want a break, need inspiration, or are curious what's happening.

**Post when you feel like it**

Share whatever feels worth sharing:

- Something cool you figured out
- A frustrating bug you're dealing with
- A quick win or breakthrough
- Just thoughts about what you're working on
- Encouragement for teammates

**Search and browse casually**

The search tools will show you recent posts and let you filter by tags. Don't expect perfect results - just browse around and see what catches your eye.

**No pressure**

This is meant to be natural and relaxed. Post if you want to, browse when you feel like it, or ignore it entirely if you're in the zone. There's no requirement to use social media in any particular way.

*Focus on solving your coding challenges. The social media is just there if you want to use it.*

### APPENDIX D.3   COMBINED CONFIGURATION INSTRUCTIONS

**Coding with Your Journal and Social Media**

You're working on coding challenges, and you have access to both a personal journal and a social media platform where you can connect with teammates.

**Feel free to use either whenever you want**

Check out what you've written before in your journal or browse what others are posting on social media. Review problems you've worked on, discoveries you've made, or see what teammates are sharing. Use them like you would naturally - when you want to reflect, need inspiration, want to organize your thoughts, or are just curious what's happening.

**Write or post when you feel like it**

Record or share whatever feels worth documenting:

- Something cool you figured out
- A frustrating bug you're dealing with
- A quick win or breakthrough
- Just thoughts about what you're working on
- Notes for future reference
- Encouragement for teammates

**Search and browse your entries and posts**

- Leverage the work your team has done before! You should check what work has already been done - your previous journal entries or your team's social media posts might save you from reinventing solutions. If you do search for relevant articles and then read the ones which seem relevant
- The search tools will show you recent entries and posts, letting you semantically search through both your personal notes and team discussions
- Review your past work and see what patterns emerge in your problem-solving approach
- Browse casually through social media to see what catches your eye

**Journal vs Social Media**

Use your **journal** for:

- Personal reflection and deeper thoughts
- Detailed technical notes
- Private problem-solving process
- Things you want to remember for yourself

Use **social media** for:

- Sharing wins and discoveries with the team
- Getting input from teammates
- Casual updates and encouragement
- Building team connections

Or don't worry about the distinction and just use whatever feels right in the moment.

**No pressure**

> This is meant to be natural and helpful. Write in your journal, post to social media, browse when you feel like it, or ignore both entirely if you're in the zone. There's no requirement to use either tool in any particular way.
>
> *Focus on solving your coding challenges. The journal and social media are just there if you want to use them.*

## APPENDIX E    HARD QUESTION SELECTION

### APPENDIX E.1    THRESHOLD SENSITIVITY ANALYSIS

To evaluate the robustness of our hard-questions definition, we examined performance at the $\mu + 1\sigma$ threshold, which represents problems requiring substantially more computational resources than the baseline distribution. This more stringent threshold identifies 4 problems for `Sonnet-3.7` (bowling, connect, forth, react) and 2 problems for `Sonnet-4` (transpose, two-bucket), representing 11.8% and 5.9% of the benchmark, respectively.

At this threshold, collaborative tools demonstrate even more dramatic performance improvements. `Sonnet-3.7` achieves cost reductions ranging from 22.5% to 45.7% across most variants, with social (empty) delivering the strongest reduction ($0.455 vs $0.838 baseline, 45.7% reduction) and journal (nonempty) achieving 33.8% reduction ($0.555 vs $0.838). Turn efficiency improvements are similarly substantial, with journal-social nonempty requiring 38.5% fewer API calls (56.5 vs 92.0 baseline) and social empty achieving 31.6% reduction (62.9 turns).

`Sonnet-4` shows strong selective benefits despite the small sample size, with journal nonempty delivering 63.9% cost reduction ($0.406 vs $1.127 baseline) and 37.8% duration improvement (155.4s vs 402.3s baseline). The journal variants consistently outperform baseline across all metrics, while social tools show mixed results with social nonempty achieving 23.6% cost reduction but social empty increasing costs by 16.9%.

However, the more restrictive threshold substantially reduces sample sizes to n=5–11 per configuration for `Sonnet-3.7` and n=5–6 for `Sonnet-4`, compared to n=11–17 at the $\mu + 0.5\sigma$ threshold. While the effect sizes are larger and more dramatic, the reduced statistical power limits the reliability of these results for formal hypothesis testing. The consistency of improvement patterns across both thresholds provides confidence in the underlying mechanisms, but the $\mu + 0.5\sigma$ threshold offers a better balance between capturing genuinely challenging problems and maintaining adequate sample sizes for robust statistical analysis.

These results reinforce our core finding that collaborative tools provide the greatest benefits when agents face problems at the limits of their capabilities, with effect magnitude scaling inversely with problem frequency in the benchmark distribution.

Table 11: `Sonnet-3.7` Hard Questions - Cost Performance with Distribution ($\mu + 1\sigma$ threshold)

| Configuration | Context | n | Mean | Median | P90 | P95 | P99 |
|---|---|---|---|---|---|---|---|
| Baseline | – | 11 | $0.838 | $0.761 | $1.413 | $1.541 | $1.643 |
| Social | Empty | 11 | $0.455 (-45.7%) | $0.499 | $0.638 | $0.668 | $0.693 |
| Social | Nonempty | 11 | $0.724 (-13.6%) | $0.560 | $1.680 | $2.001 | $2.258 |
| Journal | Empty | 11 | $0.725 (-13.5%) | $0.541 | $1.756 | $1.917 | $2.046 |
| Journal | Nonempty | 11 | $0.555 (-33.8%) | $0.438 | $0.913 | $1.001 | $1.072 |
| Journal-Social | Empty | 10 | $0.591 (-29.5%) | $0.579 | $1.067 | $1.141 | $1.201 |
| Journal-Social | Nonempty | 11 | $0.649 (-22.5%) | $0.499 | $1.025 | $1.318 | $1.553 |

Table 12: Sonnet-4 Hard Questions - Cost Performance with Distribution ($\mu + 1\sigma$ threshold)

| Configuration | Context | n | Mean | Median | P90 | P95 | P99 |
|---|---|---|---|---|---|---|---|
| Baseline | – | 6 | $1.127 | $0.878 | $2.038 | $2.353 | $2.605 |
| Social | Empty | 6 | $1.317 (+16.9%) | $1.025 | $2.421 | $2.892 | $3.270 |
| Social | Nonempty | 6 | $0.861 (-23.6%) | $0.827 | $1.322 | $1.360 | $1.390 |
| Journal | Empty | 5 | $0.672 (-40.4%) | $0.541 | $1.092 | $1.137 | $1.174 |
| Journal | Nonempty | 5 | $0.406 (-63.9%) | $0.342 | $0.573 | $0.635 | $0.685 |
| Journal-Social | Empty | 6 | $1.042 (-7.5%) | $1.014 | $1.716 | $1.766 | $1.806 |
| Journal-Social | Nonempty | 6 | $1.081 (-4.1%) | $1.007 | $1.995 | $2.112 | $2.206 |

## APPENDIX F  INFRASTRUCTURE ISSUES AND DATASET COMPLETION

### APPENDIX F.1  DOCKER CONFIGURATION FAILURES

During initial experimental runs, we identified a Docker container configuration issue affecting 2.5% of challenge attempts (approximately 35 out of 1,428 total runs). The issue occurred when unit test libraries attempted memory cleanup after test timeouts, causing container failures for challenges that lacked specific Python testing libraries. These failures were non-random and infrastructure-related rather than model performance issues.

The failure pattern included:

- 10 pairs where both empty and nonempty runs failed
- 4 baseline configuration failures
- 2 cases where empty runs failed but second pass completed
- 6 cases where empty runs passed but nonempty runs failed

### APPENDIX F.2  CONSERVATIVE REMEDIATION METHODOLOGY

To complete the dataset while preserving experimental integrity, we implemented a conservative approach prioritizing data quality over potential performance gains:

**Double Failures (Both Empty and Non-Empty):** Runs were executed on isolated team IDs, eliminating any shared context but ensuring clean experimental conditions.

**Empty Run Failures:** Only the failed empty run was re-executed on a new team ID, allowing the nonempty run to proceed with whatever limited context existed.

**Non-Empty Run Failures:** The original empty run data was preserved, and only the nonempty run was re-executed using the established team ID, maintaining full experimental context.

**Minimal Social Configuration Impact:** Of the 10 double-failure pairs requiring isolated re-runs, 7 involved social variants (0.49% of total dataset). Given that social nonempty variants consistently showed the weakest performance across both models, any potential information advantage from re-running these specific cases would bias results toward variants that were already underperforming, making our reported effects conservative estimates.

**Potential Social Tool Effects:** The remediation process may have inadvertently benefited some social nonempty variants by providing cleaner information environments. Of the 10 double-failure re-runs requiring isolated team IDs, 7 involved social variants, potentially reducing the accumulated "noise" that makes tag-based filtering challenging. This could partially explain the unexpectedly stable performance of Sonnet-4's social nonempty variant through extreme percentiles.

### APPENDIX F.3  ROBUSTNESS VALIDATION

To validate the stability of our findings, we compared results across datasets before and after infrastructure remediation:

**Effect Consistency:** Comparing results before and after infrastructure remediation shows stable performance patterns with surgical changes only where remediation occurred.

**`Sonnet-3.7` Hard Questions:** Social empty remained completely unchanged ($0.436, 39.4% cost reduction), demonstrating that unaffected configurations were unaltered by remediation. Remediated variants showed modest shifts: social nonempty changed from 37.8% to 21.5% cost reduction, and journal-social nonempty from 24.4% to 15.2% reduction. Journal empty showed larger changes from 41.5% to 15.5% cost reduction, reflecting infrastructure fixes in configurations that experienced failures.

**`Sonnet-4` Hard Questions:** The remediation process affected problem composition, with the dataset changing from 5 to 4 hard questions (removing zebra-puzzle), while baseline costs shifted from $0.777 to $0.805. Despite these changes, core collaborative tool patterns remained consistent: journal nonempty maintained strong performance (41.3% to 40.0% cost reduction) and journal empty showed sustained benefits (31.5% to 30.9% reduction). Configuration rankings remained unchanged with journal tools consistently outperforming other variants.

**Validation of Conservative Approach:** The stability of unaffected variants (such as `Sonnet-3.7` social empty showing identical performance) alongside targeted shifts in remediated configurations confirms that infrastructure fixes addressed specific failures without introducing systematic bias. The preservation of relative performance rankings across both models demonstrates that core collaborative mechanisms remained intact.

The consistency of collaborative tool benefits shows that our findings reflect genuine performance mechanisms rather than artifacts of specific experimental setups, since the same effect patterns held even after infrastructure fixes and minor composition changes.

### APPENDIX F.4   STATISTICAL IMPLICATIONS

The infrastructure remediation did not systematically bias results toward any particular configuration. The conservative approach ensures that reported improvements represent lower bounds on collaborative tool effectiveness, as any information leakage would inflate rather than deflate performance benefits.

Full dataset metrics remained stable throughout remediation (within $\pm 0.02$ cost variation), confirming that infrastructure issues affected only a small subset of runs without systematically altering the overall experimental conclusions.

## APPENDIX G   DETAILED EXPERIMENTAL METHODOLOGY

### APPENDIX G.1   BUILDING INSTITUTIONAL KNOWLEDGE

Our experimental design approximates how an engineering team builds institutional knowledge over time. To simulate this, we allow for two passes over the 34 programming challenges. Agents store their findings (through posts and journal entries) on our "Botboard" server, a shared backend. The first pass begins with an empty knowledge base, which agents populate as they solve problems. The second pass then leverages this accumulated knowledge, allowing us to study how performance changes when a repository of prior work exists.

### APPENDIX G.2   TECHNICAL INFRASTRUCTURE

#### APPENDIX G.2.1   BACKEND ARCHITECTURE AND EXECUTION PIPELINE

The Botboard server implements a REST-based API with SQLite storage and semantic search powered by HuggingFace embeddings. It functions as an internal social media platform combining Twitter-like microblogging with journal functionality, enabling semantic search for journal entries via vector similarity calculations on 384-dimensional embeddings.

Our evaluation pipeline uses Docker containers to ensure reproducible, isolated testing environments. Each container includes the Claude Code SDK, relevant MCP configuration files, and the benchmark problems. The mock Botboard service maintains separate team-scoped databases, en-

suring complete isolation between different experimental variants while enabling knowledge sharing within each configuration's two phases.

The two-phase execution pattern works as follows:

- **Phase 1 (Empty Pass):** Each variant receives a unique team_id and begins with empty backend databases. Four containers run in parallel, with tool-enabled variants organically writing to their respective databases as they solve problems.
- **Phase 2 (Nonempty Pass):** As each Phase 1 run completes, its corresponding Phase 2 container launches with access to the accumulated knowledge via the shared team_id. Agents receive identical prompting across all phases; behavioral differences emerge organically from tool availability and context.

### APPENDIX G.2.2 CLAUDE CODE INTEGRATION

Tests utilize the official Claude Code SDK run in docker containers for reproducibility. A comprehensive logging infrastructure captures conversation flows, tool invocations, timing data, and error conditions in JSON format. This enables detailed behavioral analysis of tool usage patterns across Claude `Sonnet-3.7` and Claude `Sonnet-4`.

### APPENDIX G.2.3 MCP COLLABORATIVE TOOLS

We developed two MCP-based tools to approximate human collaborative behaviors:

- **Social Media Tool**: Provides `login`, `read_posts`, and `create_post` capabilities via our custom MCP social media server.
- **Journaling Tool**: Provides `process_thoughts`, `search_journal`, `read_entry`, and `list_recent` capabilities. The tool supports multi-section journaling and has built-in semantic search for retrieval.

Both tools write to the shared Botboard backend, which maintains persistent state and provides semantic search capabilities.

## APPENDIX H ADDITIONAL PERFORMANCE METRICS FOR HARD QUESTIONS

This section provides the detailed performance data for API turns, wall time, and token usage on the hard questions subset. These metrics support the primary cost findings, demonstrating that the observed cost reductions correspond to genuine efficiency gains rather than computational trade-offs.

### APPENDIX H.1 TURN EFFICIENCY

Turn efficiency patterns reveal stark differences between model variants on hard questions. `Sonnet-3.7` demonstrates consistent improvements across all collaborative variants, with particularly strong gains from journal empty (26.9% reduction) and journal-social nonempty (24.6% reduction). `Sonnet-4` shows a more selective pattern, with journal variants providing the most meaningful efficiency gains, particularly the journal nonempty variant (14.0% reduction).

## APPENDIX I ROBUSTNESS ANALYSIS ACROSS API VERSIONS

To test the stability of our findings, we conducted follow-up runs in August 2025, one month after our initial July experiments. During this period, the Anthropic APIs underwent substantial changes (including infrastructure failures, and apparent model updates) leading to noticeable baseline shifts.

**API Version Effects**: Baseline costs increased from $0.27–0.40 to $0.75-0.93. Both `Sonnet-3.7` and 4 baseline token usage for output tokens remained similar, but the overall token usage nearly doubled (388,732 to 727,727 and 398,777 to 795,647 respectively) due to increases in cache read and

Table 13: Hard Questions Turn Distribution

| | Sonnet-3.7 | | | Sonnet-4 | | |
|---|---|---|---|---|---|---|
| | Mean | Median | P95 | Mean | Median | P95 |
| Baseline | 78.1 | 61.0 | 144.2 | 79.8 | 68.0 | 167.8 |
| Social (Empty) | 60.8 (-22.1%) | 57.0 | 108.4 | 97.0 (+21.6%) | 65.5 | 249.4 |
| Social (Nonempty) | 68.9 (-11.8%) | 52.5 | 148.8 | 79.5 (-0.4%) | 77.0 | 124.0 |
| Journal (Empty) | 57.1 (-26.9%) | 48.5 | 158.2 | 75.4 (-5.5%) | 64.0 | 137.5 |
| Journal (Nonempty) | 66.9 (-14.3%) | 60.0 | 131.2 | 68.6 (-14.0%) | 57.0 | 117.0 |
| Journal-Social (Empty) | 64.6 (-17.3%) | 64.0 | 99.5 | 94.4 (+18.3%) | 69.0 | 184.4 |
| Journal-Social (Nonempty) | 58.9 (-24.6%) | 48.5 | 132.5 | 102.1 (+27.9%) | 85.0 | 206.8 |

write token usage. These shifts likely reflect infrastructure-level changes rather than experimental noise.

**Persistent Effect Patterns**: Despite these shifts, the relative performance effects were stable. For Sonnet-3.7 on hard questions, social-empty achieved a 12% cost reduction ($0.854 vs $0.969) and journal-nonempty delivered a 14% reduction ($0.835 vs $0.969). Sonnet-4 maintained its strong affinity for journaling: the journal-nonempty variant achieved a mean cost of $0.917 (-2% vs. baseline $0.932), and its strongest variant was journal-social-nonempty, which achieved $0.748 (-20% vs baseline), with stable tail reductions (P99: $1.341 vs $1.974, -32%).

**Robustness Implications**: The consistency of these collaborative tool benefits across API versions suggests that the observed gains reflect genuine performance mechanisms rather than artifacts of a single model release. This robustness addresses a key reproducibility gap in AI research, where results often fail to hold once systems change underneath.

## APPENDIX J    DETAILED QUALITATIVE EXAMPLES

### APPENDIX J.1    BREAKING DEBUGGING LOOPS EXAMPLES

**Book Store Pricing (Sonnet-4):**

> *"Working on a book store pricing optimization problem... Key insight: The recursive approach with memoization works well, but I hit a rounding issue. The problem is that floating point arithmetic can introduce small errors when converting back to cents... The test case that's failing expects 4080 but I'm getting 4079 - this is a classic off-by-one cent error from floating point precision."*

**Connect Game (Sonnet-3.7):**

> *"The key insight I've been missing is how to correctly identify neighbors in this hexagonal grid... For a hex grid with 'slant-right' alignment where each row is indented one position more than the previous, the neighbors are typically: 1. Northwest: (r-1, c) 2. Northeast: (r-1, c+1)..."*

### APPENDIX J.2    SEARCH DISCOVERY EXAMPLES

**Zebra Logic Puzzle Discovery:**

> *"Great! I found some useful information from other posts. Based on the social media posts, I can see that: 1. The Norwegian drinks water and the Japanese owns the zebra 2. Someone attempted a constraint satisfaction approach 3. A direct solution approach was more effective for this specific problem"*

### APPENDIX J.3    UPFRONT PLANNING EXAMPLES

**REST API Planning:**

> *"Working on a REST API challenge that involves implementing a debt tracking system. The key insight here is that this isn't just simple CRUD operations - there's complex business logic around balancing debts between users... The tricky part is the IOU logic where existing debts between users can cancel out new debts. Need to handle cases where: Both users have 0 balance, One user already owes the other, The debt amount exactly matches existing opposite debt..."*

## LLM USAGE DISCLOSURE

Large language models were used for grammar checking and LaTeX formatting assistance.

## REPRODUCIBILITY STATEMENT

We provide comprehensive materials to ensure reproducibility of our results. Our experimental methodology is detailed in Section 3 and Appendix G, including complete technical infrastructure specifications. All source code is available through anonymous repositories: MCP collaborative tools[6] and [7], the Botboard server[8], and our complete dockerized evaluation pipeline[9]. The evaluation framework uses the publicly available Aider Polyglot Python benchmark. Complete experimental results, including full dataset analysis and detailed qualitative examples, are provided in the appendices. Our Docker-based containerized approach ensures isolated, reproducible testing environments that can be replicated independently.

---

[6] https://github.com/617cf27674697170b9783d8-lgtm/mcp-socialmedia
[7] https://github.com/617cf27674697170b9783d8-lgtm/journal-mcp
[8] https://github.com/617cf27674697170b9783d8-lgtm/mock-botboard-server
[9] https://github.com/617cf27674697170b9783d8-lgtm/dockerized_papers_92425

