# OpenReview forum: "AI Agents with Human-Like Collaborative Tools: Adaptive Strategies for Enhanced Problem-Solving"
_ICLR.cc/2026/Conference — Submitted to ICLR 2026_

### Official Review · Reviewer_gdZ8 · 2025-10-23

**Soundness:** 2
**Presentation:** 2
**Contribution:** 1
**Rating:** 0
**Confidence:** 4

**Summary:**

This paper studied the behavior and performance of LLM agents when provided with collaborative and automation tools. Specifically, the authors focused on the effects of adding two MCPs (social media and journaling) to Claude Code agent, and found that such enhancements improve the model’s performance while reducing costs.

**Strengths:**

Adding social media and journaling tools as MCP to the Claude code env or other agents is an interesting direction.

**Weaknesses:**

First of all, this paper lacks substantial innovation. Integrating MCPs into the Claude code environment is not a novel idea; it reads more like an engineering report. The authors implemented two MCPs within Claude code and reported their performance results. The paper fails to engage with deeper questions, such as the challenges LLMs face when using **collaborative tools** — for instance, issues like **information conflicts** or **memory management**. It details the results but lacks technical depth and falls short of the acceptance bar typically expected for ICLR.

In addition, I suggest that the authors provide a more detailed discussion of prior work on agents, agents using tools, collaborative agents. The current version cites only around 13 references, many of which are classic and general papers in the LLM field. It lacks a comprehensive review and discussion of related work specifically focused on tool use by agents, which weakens the contextual grounding of the study.

**Questions:**

Please check the comments in Weaknesses.

---

### Official Review · Reviewer_PQ8X · 2025-10-28

**Soundness:** 1
**Presentation:** 2
**Contribution:** 1
**Rating:** 2
**Confidence:** 5

**Summary:**

The paper investigated equipting an agentic model with human-like collaborative platforms such as social media and journals, to help solving challenging programming tasks. The paper first developed a synthesized socal media platform (Botboard) which combines twitter-like microblogging and journal functionality. Then, the authors conduct experiments with Claude-code using a two-pass experimental setup, where in the first pass, the agents populates the shared database, and in the second pass, the agents can access the previous populated content and try to solve the same set of tasks. Empirical results show that accessing to the Botboard reduces cost and token consumption.

**Strengths:**

- The idea of leveraging human-like collaborative platforms can be potentially a new form of communication framework for multi-agent systems, which might benefit certain tasks
- The papre writing is easy to follow

**Weaknesses:**

- The claimed contribution ("AI agents can systematically benefit from human-inspired collaboration tools") is not well supported by the experimental results:
  - The important table on completion rates (task performance) is hidden in the Appendix. And is showing that the baseline performance already saturates (99%); No benefit is shown from accessing to either journal or social in terms of task performance
  - The claimed reduction in cost/time is not convincing: (1) why accessing to an empty context can reduce computational cost? (2) is the time for MCP calls included in the computation? (3) Column titles lacks explanation: P90, P95.
 - The experimental setting has flaws: the first pass is done by the model itself and the second pass is on the same tasks, that is, the model access to its own solutions from the previous pass; thus, the gain in cost and time does not mean anything. And the performance comparison is also not fair, we should at least compare with self-refine style prompting, where the model has a second-round of verifying/modifying its initial outputs

Overall, I feel this paper is not ready for the publication, I suggest the authors to (1) find harder tasks so that the baseline performance not saturated; (2) revise the experimental setting to have more fair comparison, and include at least the self-refine baselines; (3) explain in detail what's the efficiency gain comes from for the Empty cases

**Questions:**

N/A

---

### Official Review · Reviewer_Vn14 · 2025-10-30

**Soundness:** 2
**Presentation:** 2
**Contribution:** 1
**Rating:** 2
**Confidence:** 3

**Summary:**

This work gives LLM coding agents simple human-style collaboration tools, like journals and social posts. With the freedom to use them however they want, the agents solve hard-coding tasks more efficiently. Interestingly, the models mostly write rather than read—suggesting that the main benefit comes from articulating thoughts (like rubber-duck debugging), not reusing stored knowledge.

**Strengths:**

This work tackles an under-explored but important question of whether human-like reflective tools can improve LLM agent performance, and presents an interesting empirical observation that agents naturally adopt different collaboration-style behaviors. In addition, the finding that agents benefit more from writing than reading provides a novel perspective on AI self-reflection dynamics and connects LLM behavior to known human cognitive scaffolding mechanisms.

**Weaknesses:**

1. The collaboration setup does not match the “human collaborative tools” claim.

In Figure 1, the authors say:
> (4). After the first run completes, we launch a second container (5) with MCP servers populated by previous agents’ content, allowing new agents to organically leverage accumulated knowledge

Although the paper claims to study human-like collaboration, the experimental design only executes one agent at a time. This means agents do not actually collaborate or communicate, but instead operate in a sequential memory-sharing setting. Therefore, the framing of “social collaboration” is misleading; the study evaluates self-reflection and persistent memory rather than true multi-agent collaboration.

2. Limited generalization due to reliance on Claude Code and MCP tools.

The paper only evaluates Claude Code with MCP journaling/social tools, making it unclear whether the results generalize to other agent frameworks. Since the core contribution is tool-augmented agent performance, broader validation—including open-source agent systems such as OpenHands or other LLM coding frameworks—is needed to support the generality of the claims.

3. Insufficient clarity and depth in experiments.

Although the paper reports cost, wall-time, and token usage improvements on “hard questions,” the definition of these hard questions and their source is not clearly described in the main text. This lack of transparency makes it difficult to assess the meaningfulness of the evaluation. Moreover, key performance results only appear in the appendix (e.g., Table 9 & 10), rather than the main paper. The token usage patterns reported (e.g., ~360 output tokens per turn) are also unusual and raise concerns about whether the tasks truly challenge the models. Overall, the benchmark difficulty appears limited, which weakens the strength of the main conclusion.

**Questions:**

1. How does your notion of “human-like collaboration” differ from prior multi-agent systems such as ChatDev or AgentCoder?
2. Have you evaluated performance on established software engineering benchmarks such as SWEBench-Verified?
3. Do the findings generalize to open-source agent frameworks such as OpenHands?

---

### Official Review · Reviewer_voHA · 2025-10-31

**Soundness:** 2
**Presentation:** 2
**Contribution:** 2
**Rating:** 2
**Confidence:** 4

**Summary:**

This paper demonstrates that providing LLM agents with human-like collaborative tools (journals with semantic search and social media) and the autonomy to use them spontaneously significantly enhances their problem-solving performance on challenging tasks. In rigorous coding benchmarks, agents equipped with these tools achieved 15-40% cost reductions, 12-27% fewer turns, and 12-38% faster completion times specifically on problems at their capability limits. Their behavioral analysis revealed that structured articulation was the primary driver of improvement, with agents writing 2-9x more than they read.

**Strengths:**

This paper investigates an intriguing problem — how multiple AI agents can collaborate and improve performance through networked tools, or how a model can continually learn by summarizing its experiences in microblog form. The topic is timely and relevant to advancing multi-agent systems and continual learning.

**Weaknesses:**

1. The paper does not propose concrete strategies to enhance agents’ ability to use tools, and its findings lack constructive implications.
2. As shown in Appendix B, the experiments rely on a dataset where the baseline already achieves 99% accuracy, and tool usage actually degrades performance. These counterintuitive results weaken the paper’s conclusions.
3. Only two models from the Claude family were evaluated, which is insufficient to support claims of generality.
4. Many experimental results are redundant. Metrics such as call cost, runtime, and token count are naturally correlated, yet the paper devotes excessive space to reiterating this point.

**Questions:**

1. Use a more appropriate dataset for evaluation.
2. Include models from multiple families to strengthen generality.
3. Explore scaling effects with respect to the number of agents — for instance, allowing $n$ agents to solve problems collaboratively through shared blogs, or enabling stronger agents to guide weaker ones via written posts.
4. Quotation marks are incorrectly formatted throughout the paper. The authors should use proper double quotation marks in LaTeX.

---

### Meta-Review · Area_Chair_WvAt · 2026-01-06

**Summary:**

The paper argues that giving LLM agents “human-like” collaborative tools (journals, social-style platforms) and allowing spontaneous use improves performance and efficiency on difficult programming tasks. Reviewers find the topic timely but largely agree that the evidence does not support the central claims and that the contribution is below the acceptance bar due to weak experiments, limited generality, and lack of novelty. All reviewers recommend rejection.

**Reviewer Concerns:**

* **Flawed experimental design:** Baselines already reach ~99% accuracy, leaving no headroom; tool use sometimes worsens results.
* **Unfair evaluation protocol:** Two-pass setup reuses identical tasks, confounding cost/time reductions; lacks appropriate self-refine baselines.
* **Misleading “collaboration” claim:** Only a single agent with persistent memory is used—no genuine multi-agent interaction.
* **Limited generality:** Evaluation confined to Claude Code and MCP tools; no evidence of transfer to other models or agent frameworks.
* **Low novelty/technical depth:** Seen as an engineering integration effort without new insights or methods.
* **Weak analysis:** Correlated metrics are overemphasized; “token snowball” is not rigorously studied.
* **Related work gaps:** Insufficient engagement with prior work on tool-using and collaborative agents.

**Reviewer Scores:**

The author did not provide a rebuttal, so the reviewer scores are likely to remain unchanged.

---

### Decision · Program_Chairs · 2026-01-26

Reject